# Generative Adversarial Training for MRA Image Synthesis Using Multi-Contrast MRI

**Sahin Olut, Yusuf H. Sahin, Ugur Demir, Gozde Unal**
ITU Vision Lab
Computer Engineering Department
Istanbul Technical University
{oluts, sahinyu, ugurdemir, gozde.unal}@itu.edu.tr

## Abstract

Magnetic Resonance Angiography (MRA) has become an essential MR contrast for imaging and evaluation of vascular anatomy and related diseases. MRA acquisitions are typically ordered for vascular interventions, whereas in typical scenarios, MRA sequences can be absent in the patient scans. This motivates the need for a technique that generates inexistent MRA from existing MR multi-contrast, which could be a valuable tool in retrospective subject evaluations and imaging studies. In this paper, we present a generative adversarial network (GAN) based technique to generate MRA from T1-weighted and T2-weighted MRI images, for the first time to our knowledge. To better model the representation of vessels which the MRA inherently highlights, we design a loss term dedicated to a faithful reproduction of vascularities. To that end, we incorporate steerable filter responses of the generated and reference images inside a Huber function loss term. Extending the well-established generator-discriminator architecture based on the recent PatchGAN model with the addition of steerable filter loss, the proposed steerable GAN (sGAN) method is evaluated on the large public database IXI. Experimental results show that the sGAN outperforms the baseline GAN method in terms of an overlap score with similar PSNR values, while it leads to improved visual perceptual quality.

## 1 Introduction

Due to recent improvements in hardware and software technologies of Magnetic Resonance Imaging (MRI) as well its non-invasive nature, the use of MRI has become ubiquitous in examination and evaluation of patients in hospitals. Whereas the most common MRI sequences are T1-weighted and T2-weighted MRI, which are acquired routinely in imaging protocols for studying in vivo tissue contrast and anatomical structures of interest, Non-Contrast Enhanced (NCE) time-of-flight (TOF) MR Angiography (MRA) has become established as a non-invasive modality for evaluating vascular diseases throughout intracranial, peripheral, abdominal, renal and thoracic imaging procedures [8, 17, 26]. Early detection of vessel abnormalities has vast importance in treatment of aneurysms, stenosis or evaluating risk of rupture and hemorrhage, which can be life-threatening or fatal. MRA technique has shown a high sensitivity of $95.4\%$ in detection of hemorrhage as reported in [28]. High accuracy values of $85\%$ in detection of vessel abnormalities like aneurysms using MRA are reported in [32]. In addition to NCE-MRA, Contrast-Enhanced MRA is found to improve assessment of morphological differences, while no differences were noted between NCE-MRA and CE-MRA in detection and localization of aneurysms [3]. CE-MRA is less preferred in practice due to concerns over safety of contrast agents and risks for patients as well as increased acquisition time and costs. Furthermore, when compared to the invasive modality Digital Subtraction Angiography (DSA), which is considered to be the gold standard in detection and planning of endovascular treatments, MRA is reported to present statistically similar accuracy and specificity, with a slightly reduced sensitivity

1st Conference on Medical Imaging with Deep Learning (MIDL 2018), Amsterdam, The Netherlands.

in detection of intracranial aneurysms [34]. As MRA is free of ionization exposure effects, it is a desired imaging modality for study of vasculature and its related pathologies.

In a majority of the MRI examinations, T1-weighted and T2-weighted MRI contrast sequences are the main structural imaging sequences. Unless specifically required by endovascular concerns, MRA images are often absent due to lower cost and shorter scan time considerations. When a need for a retrospective inspection of vascular structures arises, generation of the missing MRA contrast based on the available contrast could be a valuable tool in the clinical examinations.

Recent advances in machine learning, particularly emergence of convolutional neural networks (CNNs), have led to an increased interest in their application to medical image computing problems. CNNs showed a great potential in medical image analysis tasks like brain tumor segmentation and lesion detection [7, 14]. In addition to classification and segmentation related tasks, recently, deep unsupervised methods in machine learning have started to be successfully applied to reconstruction [35], image generation and synthesis problems [2]. In the training stages of such techniques, the network learns to represent the probability distributions of the available data in order to generate new or missing samples from the learned model. The main purpose of this work is to employ those image generative networks to synthesize a new MRI contrast from the other existing multi-modal MRI contrast. Our method relies on well-established idea of generative adversarial networks (GANs) [6]. The two main contributions of this paper can be summarized as follows:

- We provide a GAN framework for generation of MRA images from T1 and T2 images, for the first time to our knowledge.
- We present a dedicated new loss term, which measures fidelity of directional features of vascular structures, for an increased performance in MRA generation.

## 2   Related Works

Various methods have been proposed to generate images and/or their associated image maps. Some examples in medical image synthesis are given by Zaidi *et al.* [36] who proposed a segmentation based technique for reconstruction and refinement of MR images. Catana *et al.* suggested an atlas based method to estimate CT using MRI attenuation maps [1]. However, as the complexity of the proposed models was not capable to learn an end-to-end mapping, the performance of those models were limited [25]. Here we refer to relatively recent techniques based on convolutional deep neural networks.

**Image synthesis**, which is also termed as image-to-image translation, relied on auto-encoders or its variations like denoising auto-encoders [30], variational auto-encoders [16]. Those techniques often lead to blurry or not adequately sharpened outputs because of their classical loss measure, which is based on the standard Euclidean (L2) distance or L1 distance between the target and produced output images [23]. Generative adversarial networks (GANs) [6] address this issue by adding a discriminator to the network in order to perform adversarial training. The goal is to improve the performance of the generator in learning a realistic data distribution while trying to counterfeit the discriminator. GANs learn a function which maps a noise vector $z$ to a target image $y$. On the other hand, to produce a mapping from an image to another image, GANs can be conditioned [24]. Conditional GANs (cGANs) learn a mapping $G : x, z \rightarrow y$, by adding the input image vector $x$ to the same framework. cGANs are more suitable for image translation tasks since the conditioning vector can provide vast amount of information to the networks.

Numerous works have been published on GANs. DCGAN [27] used convolutional and fractionally strided convolutional layers (a.k.a. transposed convolution) [37] and batch normalization [10] in order to improve the performance of adversarial training. Recently, inspired from the Markov random fields [21], PatchGAN technique, which slides a window over the input image, evaluates and aggregates realness of patches, is proposed [11]. Isola *et al*'s PatchGAN method, also known as `pix2pix`, is applied to various problems in image-to-image translation such as sketches to photos, photos to maps, various maps (e.g. edges) to photos, day to night pictures and so on.

**Medical image synthesis** is currently an emerging area of interest for application of the latest image generation techniques mentioned above. Wolterink et al. [33] synthesized Computed Tomography (CT) images from T1-weighted MR images using the cyclic loss proposed in the CycleGAN technique [38]. Using 3D fully convolutional networks and available contrasts such as T1, T2, PD, T2SE images,

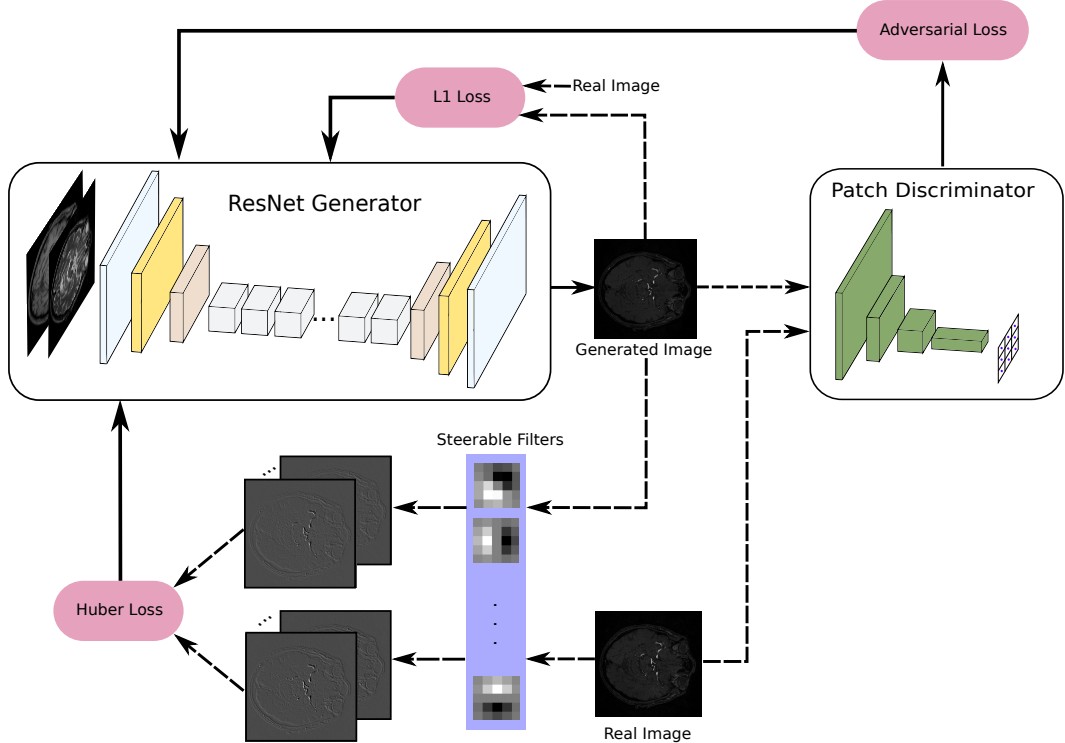

Figure 1: The sGAN architecture. ResNet generator takes concatenation of T1- and T2-weighted MR images and transforms them into an MRA image slice. The quality of the generated image is measured with three loss functions: (i) Adversarial loss generated by PatchGAN discriminator ; (ii) Reconstruction loss which evaluates pixel-wise similarity between the original and generated MRA; (iii) Steerable filter responses through a huber loss function.

Wei *et al*. [31] synthesized the associated FLAIR image contrast. Nie and Trullo *et al*. [25] proposed a context-aware technique for medical image synthesis, where they added a gradient difference as a loss term to the generator to emphasize edges. Similarly, [4] utilized CycleGAN and `pix2pix` technique in generating T1-weighted MR contrast from T2-weighted MR contrast or vice versa.

In this paper, we create a pipeline for generating MR Angiography (MRA) contrast based on multiple MRI contrast, particularly the joint T1-weighted and T2-weighted MRI using cGAN framework. As MRA imaging mainly targets visualization of vasculature, we modify the cGAN in order to adapt it to the MRA generation by elucidating vessel structures through a new loss term in its objective function. We are inspired from the **steerable filters** which arose from the idea of orientation selective excitations in the human visual cortex. Steerable filters involve a set of detectors at different orientations [18]. Orientation selective convolution kernels are utilized in various image processing tasks such as enhancement, feature extraction, and vesselness filtering [5, 19]. In our work, the addition of the steerable filter responses to the cGAN objective tailors the generator features to both reveal and stay faithful to vessel-like structures, as will be demonstrated.

## 3 Method

The proposed method for generating a mapping from T1- and T2-weighted MRI to MRA images, which is named as steerable filter GAN (sGAN), is illustrated in Figure 1. The generator and the discriminator networks are conditioned on T1- and T2-weighted MRI, which are fed to the network as two channels of the input. The details of the proposed architecture are described next.

## 3.1 Generator network

An encoder-decoder type generator network with residual blocks [9] that is similar to the architecture introduced in [13] is adopted in sGAN. Our network consists of 3 down-sampling layers with strided convolutions of stride 2, which is followed by 9 residual blocks. In residual blocks, the channel size of input and output are the same. At the up-sampling part, 3 convolutions with fractional strides are utilized. All convolutional layers except the last up-sampling layer are followed by batch normalization [10] and ReLU activation. In the last layer, tanh activation without a normalization layer is used.

## 3.2 Discriminator network

In a GAN setting, the adversarial loss obtained from the discriminator network D forces the generator network G to produce sharper images, while it updates itself to distinguish real images from synthetic images. As shown in [11], the Markovian discriminator (PatchGAN) architecture leads to more refined outputs with detailed texture as both the input is divided to patches and the network evaluates patches instead of the whole image at once. Our discriminator architecture consists of 3 down-sampling layers with strides of 2 which are followed by 2 convolutional layers. In the discriminator network, convolutional layers are followed by batch normalization and LReLU [22] activation.

## 3.3 Objective Functions

In sGAN, we employ three different objective functions to optimize parameters of our network. **Adversarial loss**, which is based on the original cGAN framework, is defined as follows:

$$\mathcal{L}_{GAN}(G, D) = \mathbb{E}_{x,y}[\log D(x, y)] + \mathbb{E}_x[\log(1 - D(x, G(x)))] \tag{1}$$

where $G$ is generator network and $D$ is discriminator network, $x$ is the two channel input consisting of T1-weighted and T2-weighted MR images, $G(x)$ is the generated MRA image, and $y$ is the reference (target) MRA image, respectively. We utilize the PatchGAN approach, where similarly, the adversarial loss evaluates whether its input patch is real or synthetically generated [11]. The generator is trained with $\mathcal{L}_{adv}$ which consists of the second term in Equation 1.

**Reconstruction loss** helps the network to capture global appearance characteristics as well as relatively coarse features of the target image in the reconstructed image. For that purpose, we utilize the $L1$ distance, which is calculated as the absolute differences between the synthesized output and the target images:

$$\mathcal{L}_{rec} = ||y - \hat{y}||_1 \tag{2}$$

where $y$ is the target, $\hat{y} = G(x)$ is the produced output.

**Steerable filter response loss**

As MR Angiography specifically targets imaging of the vascular anatomy, faithful reproduction of vessel structures is of utmost importance. Recently, it is shown that variants of GANs with additional loss terms geared towards the applied problem can achieve improved performance compared to the conventional GANs [20]. Hence, we design a loss term that emphasizes vesselness properties in the output images. We resort to steerable filters that are orientation selective convolutional kernels to extract directional image features tuned to vasculature. In order to increase the focus of the generator towards vessels, we propose the following dedicated loss term which further incorporates a Huber function loss involving a combination of an L1 and L2 distance between steerable filter responses of the target image and the synthesized output:

$$\mathcal{L}_{steer} = \frac{1}{K} \sum_{k=1}^{K} \rho(f_k * y, f_k * \hat{y}) \tag{3}$$

where $*$ denotes the convolution operator, $K$ is the number of filters, $f_k$ is the $k^{th}$ steerable filter kernel. The Huber function with its parameter set to unity is defined as:

$$\rho(x, y) = \begin{cases} (x - y)^2 * 0.5 & if \ |x - y| \leq 1 \\ |x - y| - 0.5 & otherwise \end{cases}$$

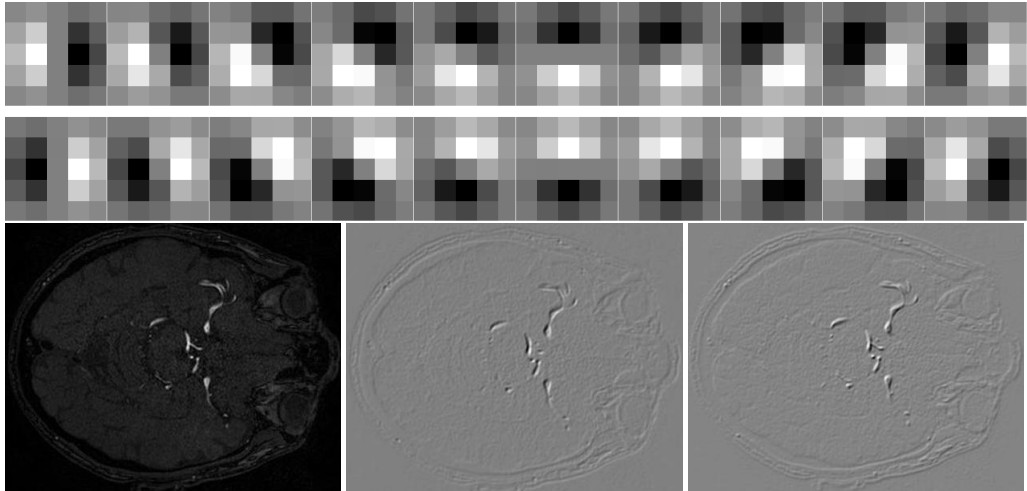

Figure 2: Top two rows: Generated steerable filter kernel weights ($5 \times 5$); Bottom row: two examples of steerable filter responses ($k = 7, 18$) to the input MRA image on the left.

Figure 2 depicts the $K = 20$ steerable filters of size $5 \times 5$. We also show sample filter responses to an MRA image to illustrate different characteristics highlighted by the steerable filters.

In the sGAN setting, the overall objective is defined as follows:

$$\mathcal{L} = \lambda_1 \mathcal{L}_{adv} + \lambda_2 \mathcal{L}_{rec} + \lambda_3 \mathcal{L}_{steer} \tag{4}$$

where $\mathcal{L}_{adv}, \mathcal{L}_{rec}, \mathcal{L}_{steer}$ refer to Equations 1,2,3, respectively, with corresponding weights $\lambda_1, \lambda_2, \lambda_3$.

## 4 Experiments and Results

### 4.1 Dataset and Experiment Settings

We use IXI dataset[1] which includes 578 MR Angiography and T1- and T2-weighted images. As the images are not registered, we used rigid registration provided by FSL software [29] to register T1- and T2-weighted to MRA. In training, we utilize 400 MRA volumes of size 512 x 512 x 100, and randomly selected additional 40 volumes are used for testing.

The MRA scans in the dataset have higher spatial resolution in the axial plan, therefore, the sGAN architecture is slice-based. The image slices in a volume are normalized according to the mean and standard deviation of the whole brain. The sGAN model is trained for 50 epochs and learning rate is linearly decreased after 30 epochs. The parameters used in the model are: learning rate 0.0002, loss term constants $\lambda_1 = 0.005$, $\lambda_2 = 0.8$ and $\lambda_3 = 0.145$ in Equation 4. Adam [15] optimizer is used with $\beta_1 = 0.5$ and $\beta_2 = 0.999$. PyTorch[2] framework is used in all of our experiments which are run on NVIDIA$^{TM}$ Tesla K80 GPUs. We trained both models for a week. A feedforward pass for each brain generation takes about 10 seconds.

### 4.2 Evaluation metrics

We utilize two different measures for performance evaluation. First one is the peak signal-to-noise ratio (PSNR) which is defined by

$$PSNR = 10 \log_{10} \frac{(\max y)^2}{\frac{1}{n} \sum_i^n (y_i - \hat{y}_i)^2},$$

where n is the number of pixels in an image. The PSNR is calculated between the original MRA and the generated MRA images.

---

[1]http://brain-development.org/ixi-dataset

[2]http://pytorch.org

In the MRA modality generation, it is important to synthesize vessel structures correctly. We utilize Dice score as the second measure in order to highlight the fidelity of the captured vascular anatomy in the synthesized MRA images. The Dice score is defined by

$$Dice(y, \hat{y}) = \frac{2|y \cap \hat{y}|}{|y| + |\hat{y}|}.$$

In order to calculate the Dice score, the segmentation maps are produced by an automatic vessel segmentation algorithm presented in [12] over both the original MRA images and the generated MRA images using the same set of parameters in the segmentation method. As reference vascular segmentation maps are not available in the IXI dataset, we calculated the Dice score as the overlap between the vasculature segmented on the original images and the vasculature segmented on the generated images.

### 4.3 Quantitative Results

To our knowledge, no previous works attempted synthetic MRA generation. To evaluate our results, we compare the generated MRA images corresponding to the baseline, which is the PatchGAN with ResNet architecture against the sGAN, which is the baseline with added steerable loss term. The PSNR and Dice scores are tabulated in Table 1.

| Method | PSNR (dB) | Dice Score (%) |
|---|---|---|
| Baseline: $\mathcal{L}_{adv} + \mathcal{L}_{rec}$ | 29.40 | 74.8 |
| sGAN: $\mathcal{L}_{adv} + \mathcal{L}_{rec} + \mathcal{L}_{steer}$ | **29.51** | **76.8** |

Table 1: Performance measures (mean PSNR and mean Dice scores) on the test set: first row corresponds to the baseline PatchGAN; second row shows the sGAN results.

### 4.4 Qualitative Results

We show sample visual results of representative slices in Figure 3. Sample 3D visual results are given as surface renderings of segmentation maps in Figure 4.

## 5 Discussion and Conclusion

MRA is based on different relaxation properties of moving spins in flowing blood inside vessels, compared to those of static spins found in other tissue. The presented sGAN method is a data-driven approach to generation of MRA contrast, from the multi-contrast T1- and T2-weighted MRI, which are based on spin-lattice and spin-spin relaxation effects. It is possible to include other available MR contrast in patient scans such as Proton Density, FLAIR, and so on, as additional input channels to the sGAN network.

The sGAN relies on the recent popular PatchGAN framework as the baseline. In the adaptation of the baseline method to MRA generation, the steerable-filter response based loss term included in the sGAN method highlights the directional features of vessel structures. This leads to an enhanced smoothing along vessels while improving their continuity. This is demonstrated qualitatively through visual inspection. In quantitative evaluations, the sGAN performs similarly with a slight increase (statistically insignificant) in PSNR values compared to those of the baseline. However, it is well-known that PSNR measure does not necessarily correspond to perceptual quality in image evaluations [20, 23]. In terms of the vascular segmentation maps extracted from the generated MRAs and the original MRA, the sGAN improves the overlap scores by 2% against the baseline. This is a desirable output, as the MRA targets imaging of vascular anatomy.

The presented sGAN method involves 2D slice generation. This choice is based on the native axial acquisition plan of the MRA sequences, hence the generated MRA has expectedly higher resolution in the axial plane. Our future work includes extension of sGAN to a fully 3D architecture. Making use of 3D neighborhood information, both in generator networks and 3D steerable filter responses is expected to increase the continuity of vessels in 3D.

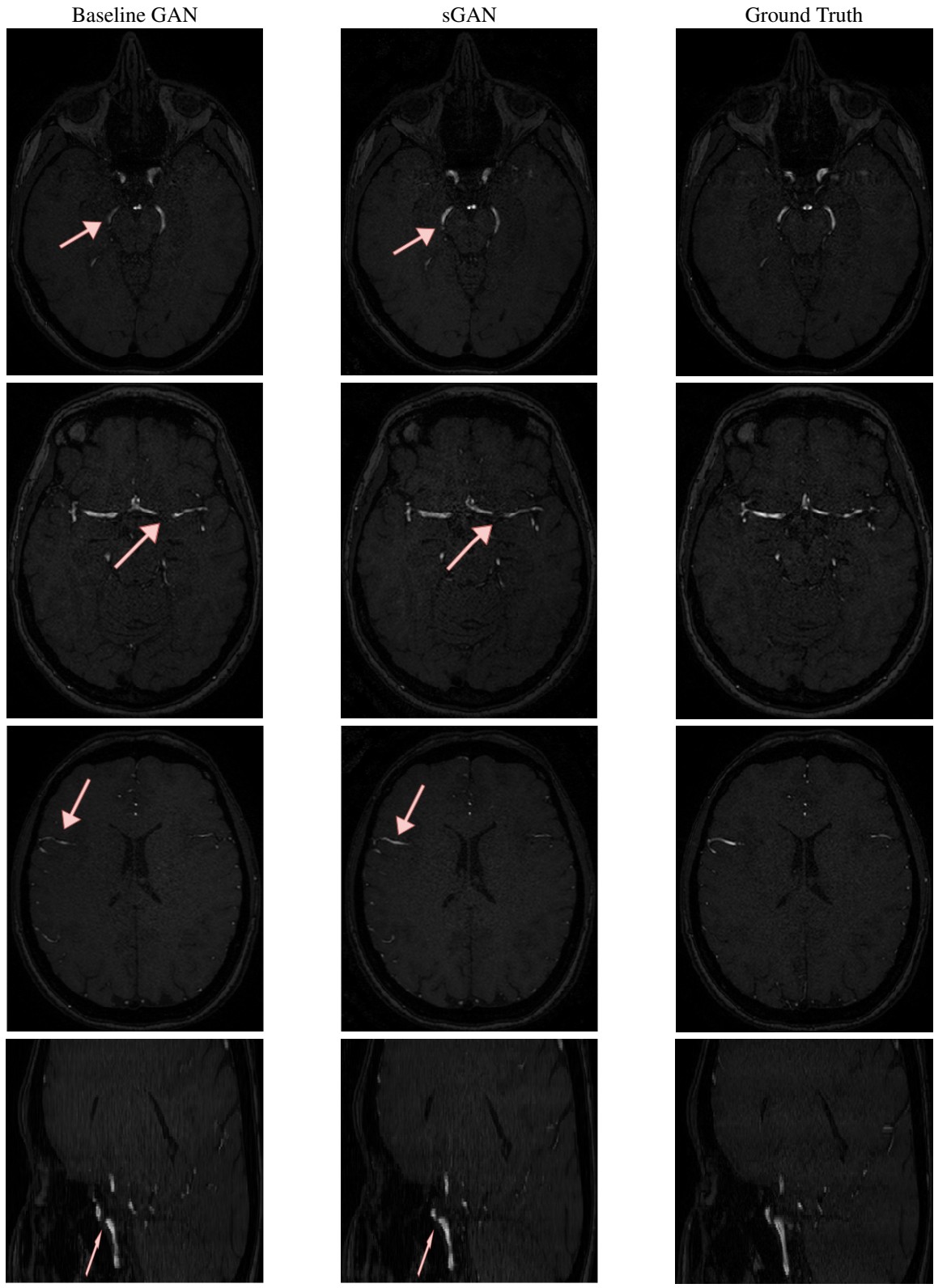

Figure 3: Visual comparison of generated 2D MRA axial slices to the original MRA slices by both the baseline and the sGAN methods. The last row shows a sagittal slice.

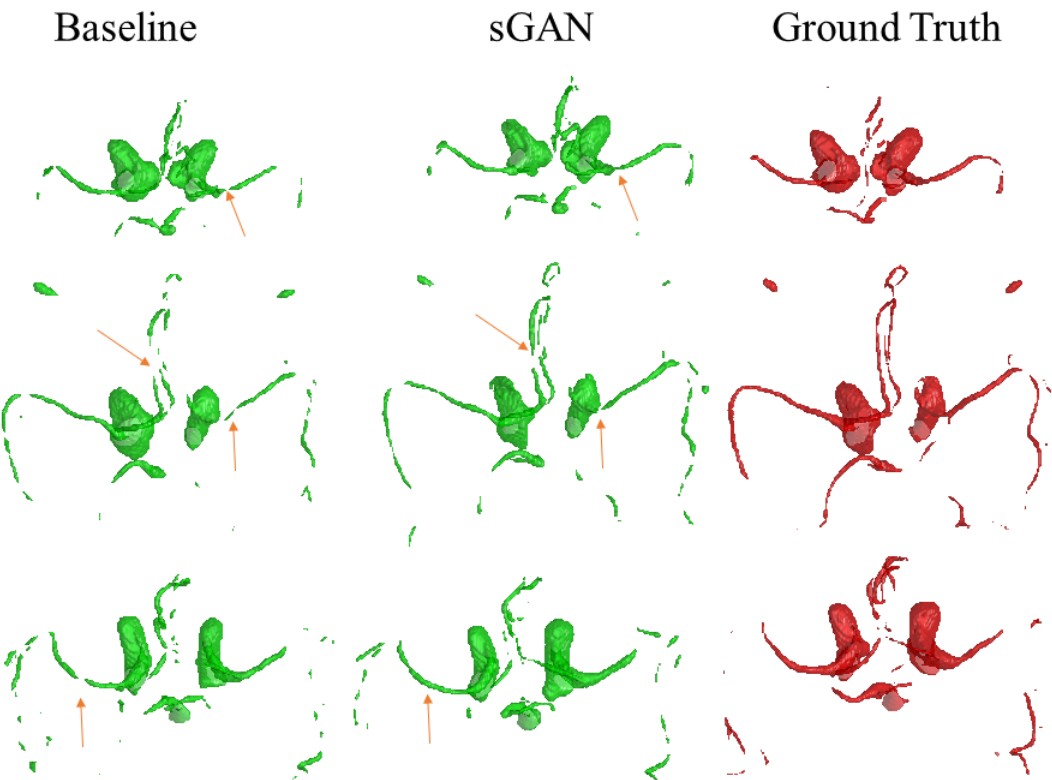

Figure 4: Visual comparison of segmentation maps over generated MRA to those over the original MRA in surface rendering format using both the baseline and the sGAN methods.

The proposed sGAN has the potential to be useful in retrospective studies of existing MR image databases that lack MRA contrast. Furthermore, after extensive validation, it could lead to cost and time effectiveness where it is needed, by construction of the MRA based on relatively more common sequences such as T1- and T2-weighted MR contrast.

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
