# OpenReview forum: "Generative Adversarial Training for MRA Image Synthesis Using Multi-Contrast MRI"
_MIDL.amsterdam/2018/Conference — MIDL 2018 Poster_

### Review · AnonReviewer1 · 2018-05-02
**The authors present a novel work on contrast transformation from existing T1- & T2-weighted MR images to images showing angiographic contrast. An extended (steerable) generative adversarial network (sGAN) was trained to generate non-measured contrast. The key development is the design of a steerable filter loss accounting for vascular-specific geometry. The basic idea is clearly described, but there are shortcomings on methods, i.e. experimental details, as well as quantification of results.**

**Rating:** 4
**Confidence:** 2

**Review:**

Assuming, that the main idea is to present a technique to retrospectively generate non-measured MRA images, this manuscript provides a useful and relevant approach for future clinical application. However, several shortcomings should be addressed more detailed as mentioned point by point below.

Section 3), your methods are, in general, described sufficiently except the remarks mentioned here. However, I suggest a general revision of this section, i.e. formula and text for adversarial loss (3.3). It should be pointed out more clearly which loss is used for PatchGAN.

Section 4), used data: Why did you not use all available datasets (400 training + 40 testing vs. 578 images in ixi-dataset)?
MR specific details: The information given on the used MR images are rare. Although, referring to the ixi-dataset may provide these information direct mentioning in the manuscript would ease readability and complete scientific standard. Missing information are e.g. scanner-system used, i.e. field strength, equality of the protocols. However, assuming a target audience with background in informatics, it may be sufficient to refer to literature and exclude MR specific details nearly completely as done in the manuscript.
Note: Information on http://brain-development.org/ixi-dataset/ are limited and no dicom tags are available, because not explicitly stated I assume T2-weighted images were acquired with a gradient-echo type of sequence. This would mean that your ‘true’ contrast is actually not T2, but rather T2* (http://www.mri-q.com/t2-vs-t2.html).

Section 4), are both networks trained for the same number of epochs? Details for training of sGAN are given, but for PatchGAN these are less detailed.

Section 4.2) and 4.3), a comparison between to networks (sGAN, PatchGAN) is done and quantified by some metric (PSNR, Dice score). Maybe an additional (more perceptual) metric and/or L1 or L2 between original and generated MRA images may be helpful. Are your results only for a single subject (and slices) or is this a statistical result for several subjects, i.e. all 40 test datasets?

Presentation of results: Shown images (Fig. 3, Fig. 4) seem to be from a single subject. Why did you not show at least two/more subjects? This would circumstantiate the performance of your sGAN, with only one subject it rather raises doubts. The caption should contain this information if not clearly labelled in the figure. The text in 4.4 is fairly short to what is shown in the figures.

Section 5), first sentence “MRA is…” is physically incorrect and should not be used as a short introduction to the principles of MRA. If you want to give some basic information on the principles of MRA these should be given in the introduction and be more extensive. Note: I couldn’t find very specific information on the MRA-sequence used for ixi-dataset on their homepage, but the type of sequence determines the physical basic of ‘how the contrast is produced’.

Section 5), Discussion and Conclusion: In general your discussion is fairly short and I would suggest extensions on different aspects.
- The potential clinical value could be addressed in more detail.
- If it is possible to include PD or FLAIR images into the training why didn’t you do this? Or is this an assumption only? Please, clarify. If an assumption, why is this beneficial.

Style:
- Labelling of figures, arrow in Fig 3 is not described in the caption
- Bold font in text (“steerable filters”) in contrast to heading
- Section 4.3), first sentence “ To our knowledge, …” has no meaning in this section, should be 	removed.
- Section 4.2), equations are not numbered

Language:
- The use of abbreviations is inconsistent, abbreviations should be introduced once and then used without exception (“…estimate CT”, “T2SE”, “FLAIR”, “MRA” multiple introductions, “PD” used but not introduced (only named) before).
- Please, double-check your English (3.2 Discriminator network, 4.1 “…models for a (one) week.”).
- Section 3.2, “…sharper images…” may the term “…more realistic images…” would be more suitable
- MRI stand for Magnetic Resonance Imaging, in e.g. section 3) “…T2-weighted MRI to…” and “… T2-weighted MRI, which…” you are using it as Magnetic Resonance Image, you should use “MR image”.
- “Axial plane” instead of “axial plan” is more common at least in my option, but this is of course only a minor remark.


**Special Issue:**

Yes

---

### Review · AnonReviewer2 · 2018-05-09
**GAN with new loss function for image synthesis**

**Rating:** 4
**Confidence:** 3

**Review:**

The paper presents a novel approach to image synthesis using the GAN framework with application specific loss functions.

The methodology and problem formulation is very interesting, and the design choices are sensible. The steerable filter response loss seems to be the main contribution, and this seem to work well for the task of MRA synthesis and improved reconstruction of vascular anatomy.

Technically, the paper is sound and likely to be of interest to a larger audience.

I cannot really judge the practical value of this approach, as I find it difficult to believe that image synthesis (in general) can yield images that can be clinically useful. GANs in particular may suffer from hallucination, which is a highly questionable characteristic in medical imaging. Generally, the basic assumption must hold that there exists a nonlinear relationship between the input and output, and otherwise the problem is ill-posed and not solvable with via function approximation (aka neural networks).

The authors might want to consider a different performance measure than Dice. Something more suitable to quantify the reconstruction of tubular structures/centerlines could be more meaningful and might work in favor of the proposed approach.

**Special Issue:**

Yes

---

### Review · AnonReviewer3 · 2018-05-09
**Review of Generative Adversarial Training for MRA Image Synthesis Using Multi-Contrast MRI**

**Rating:** 3
**Confidence:** 3

**Review:**

The authors presented a generative adversarial network (GAN) based technique to generate MRA from T1-weighted and T2-weighted MRI images. This topic looks very important in clinical applications.

1. What's the original resolution of T1, T2 images?
2. Please shows SDs in Table 1. In addition, shows statistical comparison between results.
3. How do you think about false positive and false negatives of this approaches. Is it clinical applicable?

**Special Issue:**

No

---

### Decision · Program_Chairs · 2018-05-15
**Paper86 Acceptance Decision**

Poster